# Simplifying Knowledge Transfer in Pretrained Models

**Siddharth Jain**                                    *siddharth.ja@research.iiit.ac.in*
*Center for Visual Information Technology*
*International Institute of Information Technology, Hyderabad*

**Shyamgopal Karthik**                                *shyamgopal.karthik@uni-tuebingen.de*
*University of Tübingen*

**Vineet Gandhi**                                     *vgandhi@iiit.ac.in*
*Center for Visual Information Technology*
*International Institute of Information Technology, Hyderabad*

**Reviewed on OpenReview:** *https://openreview.net/forum?id=eQ9AVtDaP3*

## Abstract

Pretrained models are ubiquitous in the current deep learning landscape, offering strong results on a broad range of tasks. Recent works have shown that models differing in various design choices exhibit categorically diverse generalization behavior, resulting in one model grasping distinct data-specific insights unavailable to the other. In this paper, we propose to leverage large publicly available model repositories as an auxiliary source of model improvements. We introduce a data partitioning strategy where pretrained models autonomously adopt either the role of a student, seeking knowledge, or that of a teacher, imparting knowledge. Experiments across various tasks demonstrate the effectiveness of our proposed approach. In image classification, we improved the performance of ViT-B by approximately 1.4% through bidirectional knowledge transfer with ViT-T. For semantic segmentation, our method boosted all evaluation metrics by enabling knowledge transfer both within and across backbone architectures. In video saliency prediction, our approach achieved a new state-of-the-art. We further extend our approach to knowledge transfer between multiple models, leading to considerable performance improvements for all model participants. The code is available at: https://github.com/Syd-J/Bi-KD

## 1 Introduction

Knowledge Distillation (KD) (Buciluǎ et al., 2006; Hinton et al., 2015; Beyer et al., 2022) intends to transfer knowledge from a large 'teacher' model to a smaller 'student' model. Traditional KD methods utilize the predictions from the pretrained teacher model to supervise the training of the student model, encouraging it to generalize better than if it were trained from scratch alone. However, vanilla KD is a two-stage process that begins with training a teacher model and then freezing it to distill knowledge into the student model, meaning that the knowledge can only be transferred from the teacher to the student. Online Knowledge Distillation methods (Zhang et al., 2018; Guo et al., 2020) overcome this limitation by adopting a one-stage training process, jointly training a set of student models that learn from each other in a peer-teaching manner.

Although online KD methods employ a single stage training process, they distill knowledge into untrained student models. Transferring knowledge in this way neglects the existence of complementary knowledge between pretrained models, a factor that, if considered, can enhance generalization (Gontijo-Lopes et al., 2022; Roth et al., 2024). Various design choices such as hyperparameters, model architecture, optimization strategies, and pretraining dataset shape the semantic knowledge a model acquires (Bouthillier et al., 2021;

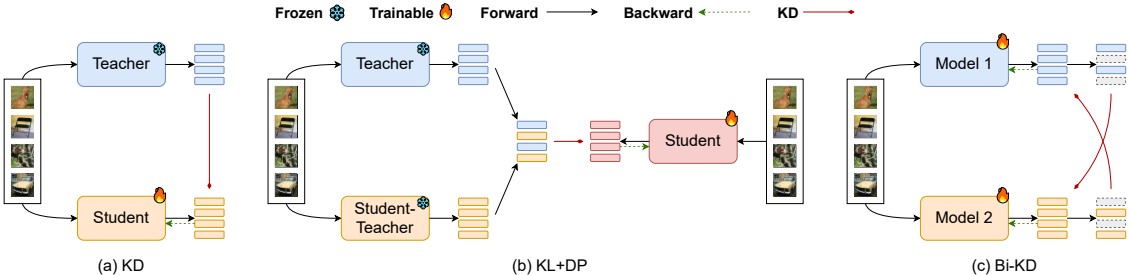

Figure 1: The figure illustrates the training process for a given batch of samples. (a) KD (Hinton et al., 2015) utilizes a pretrained teacher model and trains a student model to mimic the teachers predictions on each sample. (b) KL+DP (Roth et al., 2024) employs a frozen teacher and the original frozen student (called student-teacher) to jointly guide the training of the student model. They partition the dataset into samples where learning from the teacher is desired and ones where knowledge from the original frozen student should be retained. (c) In Bi-KD (ours), both models are trainable and learn from each other on every sample, resulting in bidirectional knowledge transfer. Different from KD and KL+DP, we are able to improve both models simultaneously.

Wagner et al., 2022). Gontijo-Lopes et al. (2022) show that model pairs with diverging training methodologies produce increasingly uncorrelated errors. Even low-accuracy models may capture some data-specific insights that high-accuracy models might overlook. Roth et al. (2024) capitalize on this finding by transferring complementary knowledge between any pretrained teacher and student model pair. They propose a data partitioning strategy that divides the dataset into instances where knowledge transfer from a teacher is beneficial and those where retaining the student's behavior is preferred. However, their approach operates in a unidirectional manner, wherein the teacher remains fixed after pretraining and the student is trained to minimize the teacher-student gap. This contrasts with the real-world teacher-student dynamic, in which a teacher continuously improves their knowledge and teaching skills through ongoing interactions with the student (Cornelius-White, 2007; Wright, 2011).

Transferring knowledge among a group of pretrained models lets each one benefit from the unique strengths and insights that the others have already developed. They can complement each other's weaknesses and build more robust, generalized representations. When models are trained together, they iteratively update their predictions, effectively "teaching" each other. This simultaneous updating can lead to a performance boost that none of the models might achieve if trained independently (Zhang et al., 2018; Zhu et al., 2018; Guo et al., 2020).

To this end, we propose a simple method for bidirectional knowledge transfer between pretrained models, a challenging task due to their prior training. Building on the data partition strategy introduced by Roth et al. (2024), we dynamically assign the teacher role to the model with the highest prediction confidence for the ground-truth class, allowing it to transfer knowledge to the other model. Unlike their fixed partitioning approach, our confidence-based data partitioning evolves as the models improve, adapting throughout the training process (as illustrated in Figure 1). Despite the apparent simplicity of our method, it leads to consistent performance improvements for both the models within a single training stage.

We validated our approach through extensive experiments on models with diverse architectures, performance levels, sizes, and training objectives. Our method was tested across multiple tasks, including image classification, semantic segmentation, and video saliency prediction, and was further extended to enable parallel knowledge transfer among multiple models. In all cases, we observed consistent performance improvements.

Overall, we make the following contributions:

- We demonstrate the ability for bidirectional knowledge transfer in pretrained models. Specifically, we show that knowledge can be transferred across both models simultaneously.

- We provide experiments across ImageNet classification, semantic segmentation, and video saliency prediction, where we observe consistent improvements for all participating models. In particular, our method sets a new state-of-the-art in video saliency prediction.

- We establish that our framework seamlessly extends to concurrent knowledge transfer across multiple models, thereby progressively enhancing the performance of each individual model as additional models are integrated.

## 2 Related Work

**Knowledge Distillation** (KD), pioneered by Buciluǎ et al. (2006), aimed to compress large teacher models into smaller student models by aligning their soft target distributions. Hinton et al. (2015) refined this approach by incorporating temperature scaling to minimize the difference between the softened class probabilities of the teacher and student models. Beyer et al. (2022) further highlighted the importance of consistent image augmentations and extended training schedules for effective KD.

Building on these ideas, recent works have explored transferring knowledge beyond just output probabilities. Romero et al. (2014) proposed aligning the intermediate feature representations between the teacher and the student models, while Zagoruyko & Komodakis (2017) train a student to imitate the attention maps of teacher networks. Park et al. (2019) introduced a method that preserves the structural relationship between the outputs using distance-wise and angle-wise losses. However, these methods often require careful layer selection and loss balancing (Yun et al., 2019), making them highly dependent on specific network architectures. To address this, Srinivas & Fleuret (2018) proposed matching the Jacobian of network outputs, which has a dimension independent of the model's architecture.

The work most similar to ours, proposed by Roth et al. (2024), introduced a data partitioning strategy for knowledge transfer among pretrained models. However, their approach uses a fixed data partitioning strategy during training, and the teacher model is not optimized specifically to guide the student. In contrast, our work introduces an evolving data partitioning strategy, which enables continuous improvements in the models through repeated interactions.

**Online Knowledge Distillation** treats every network as a student and trains them simultaneously from scratch. DML (Zhang et al., 2018) enables peer student models to learn from each other's predictions using a combination of cross-entropy and distillation losses. Anil et al. (2018) extend this concept to large-scale distributed neural networks, accelerating training by updating models concurrently.

Other approaches (Song & Chai, 2018; Zhu et al., 2018) involve designing multiple branch classifiers that are trained together. However, these methods are inflexible as they force networks to share lower layers, restricting knowledge transfer to only the upper layers within a single model. Chen et al. (2020) incorporate a self-attention mechanism to assess the similarity between network groups, enhancing peer diversity to create a more effective leader. Similarly, KDCL (Guo et al., 2020) introduces an ensemble of logits, where the optimal weight distribution is determined using a Lagrange multiplier to minimize generalization error.

More recently, Wu & Gong (2021) introduce an extra temporal mean network for each peer, assigning it the teacher role. Li & Jin (2022) propose a proxy teacher that updates its weights based on predictions from the original teacher model, enabling bidirectional distillation with the student model. While these methods optimize the teacher model for distillation, they transfer knowledge between untrained models, disregarding the presence of complementary knowledge between pretrained models. Livanos et al. (2024) address this by transferring knowledge between trained models. They dynamically assign the teacher role to a model that correctly classifies an instance while others fail. The teacher then generates a counterfactual instance for each correct prediction, adding it to the training set of incorrect models, which are subsequently retrained. However, this approach requires multiple training stages, making it computationally inefficient.

In contrast, our proposed method improves every pretrained model involved in KD within a single training stage, leading to a more efficient and effective learning process.

**Multi-teacher Knowledge Distillation.** KD can naturally be extended to learning from multiple pretrained teachers. Fukuda et al. (2017) combine the distillation framework with a data augmentation strategy

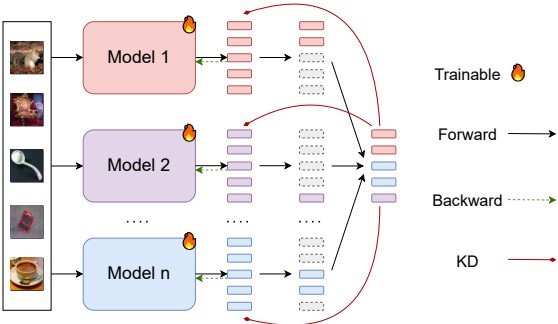

Figure 2: Generalization of our proposed method for transferring knowledge between multiple pretrained models. For each sample, we select a teacher model that guides the training of all other models. The teacher is selected based on the highest prediction probability, or the lowest loss, corresponding to the ground-truth. This data partitioning strategy enables every model participating in the knowledge transfer to learn from each others strengths and address their weaknesses, resulting in all models improving within a single training stage.

by creating multiple copies of the data with the corresponding soft output targets from multiple teachers. You et al. (2017) further extend this approach by incorporating multiple teacher networks in the intermediate layers, considering the dissimilarity between intermediate representations of different examples. Luo et al. (2019) propose a common feature learning scheme, in which the features of all teachers are transformed into a common space, and the student is required to imitate them all to amalgamate the intact knowledge. Instead of treating all teacher models equally, some works (Liu et al., 2020; Yuan et al., 2021) dynamically assign weights to teacher models for different training instances and optimize the performance of the student model.

Although these works utilize predictions from multiple teacher models to reduce variance in network outputs, they fail to optimize teacher networks by considering the complementary knowledge between them. Our proposed method allows every model to benefit from each other's strengths and complement their weaknesses. This results in consistent performance improvements for all models in a single training stage, which ultimately leads to more robust predictions.

## 3 Bidirectional Knowledge Transfer

In this section, we first explain the traditional KD approach in Section 3.1, then introduce our proposed method for bidirectional knowledge transfer between two pretrained models in Section 3.2. Section 3.3 explains the extension of our method to dense tasks, namely semantic segmentation, and video saliency prediction, and finally, Section 3.4 highlights our approach for multidirectional knowledge transfer among multiple pretrained models.

### 3.1 Preliminaries

KD aims to improve the performance of the student network by leveraging the predictions of a teacher network as supervision. Hinton et al. (2015) propose minimizing the Kullback-Leibler (KL) divergence between the soft targets of the teacher and student models. The distillation loss is formulated as:

$$\mathcal{L}_{KL_{1,2}} = \frac{T^2}{N} \sum_{i=1}^{N} \text{KL}\left[\sigma(\mathbf{z}_{1,i}/T), \sigma(\mathbf{z}_{2,i}/T)\right] \tag{1}$$

where $T$ represents the temperature parameter, $N$ denotes the batch size, and $\sigma(\mathbf{z}_1)$ and $\sigma(\mathbf{z}_2)$ correspond to class probabilities of student and teacher predictions, respectively. We use Equation 1 along with task-specific loss as our overall loss function.

---

**Algorithm 1:** Bi-KD

---

**Input:** Training set $\mathcal{X}$, label set $\mathcal{Y}$, learning rate $\eta$, epochs $T_{\max}$, iterations $N_{\max}$, models $f_1$ and $f_2$ parameterized by $\theta_1$ and $\theta_2$ respectively

**1** **for** $T = 1$ **to** $T_{\max}$ **do**

**2**      Shuffle training set $\mathcal{X}$;

**3**      **for** $N = 1$ **to** $N_{\max}$ **do**

**4**          Fetch mini-batch $x$ from $\mathcal{X}$;

**5**          Compute output logits $\mathbf{z}_1 = f_1(x; \theta_1)$ and $\mathbf{z}_2 = f_2(x; \theta_2)$;

**6**          Obtain data mask $m_1 = \mathbb{I}\big[\sigma(\mathbf{z}_1)_{gt} > \sigma(\mathbf{z}_2)_{gt}\big]$ ;     `// samples where` $f_1$ `acts as a teacher`

**7**          Obtain data mask $m_2 = \mathbb{I}\big[\sigma(\mathbf{z}_1)_{gt} \leq \sigma(\mathbf{z}_2)_{gt}\big]$ ;     `// samples where` $f_2$ `acts as a teacher`

**8**          Get the distillation loss $\mathcal{L}_{\mathrm{dist}} = m_1 \cdot \mathcal{L}_{KL_{2,1}} + m_2 \cdot \mathcal{L}_{KL_{1,2}}$;

**9**          Get the cross-entropy losses $\mathcal{L}_{\mathrm{ce}_1} = \mathrm{CE}(f_1(x; \theta_1), \mathcal{Y})$ and $\mathcal{L}_{\mathrm{ce}_2} = \mathrm{CE}(f_2(x; \theta_2), \mathcal{Y})$;

**10**         Compute overall loss $\mathcal{L} = \mathcal{L}_{\mathrm{ce}_1} + \mathcal{L}_{\mathrm{ce}_2} + \mathcal{L}_{\mathrm{dist}}$;

**11**         Update $\theta_1' \leftarrow \theta_1 - \eta \frac{\partial \mathcal{L}}{\partial \theta_1}$;

**12**         Update $\theta_2' \leftarrow \theta_2 - \eta \frac{\partial \mathcal{L}}{\partial \theta_2}$;

**Output:** Trained models $f_1(\theta_1')$ and $f_2(\theta_2')$

---

### 3.2 Formulation

Given a training data batch $\mathcal{D} = (\mathcal{X}, \mathcal{Y})$ of $N$ samples, where $\mathcal{X} = \{x_i\}_{i=1}^N$ and $\mathcal{Y} = \{y_i\}_{i=1}^N$ represent the inputs and corresponding labels for $C$ classes, we consider two models $f_1$ and $f_2$ parameterized by $\theta_1$ and $\theta_2$ respectively. Their output logits are computed as follows:

$$\mathbf{z}_1 = f_1(\mathcal{X}; \theta_1), \quad \mathbf{z}_2 = f_2(\mathcal{X}; \theta_2) \tag{2}$$

For each sample, we dynamically assign the teacher role to the model with the highest prediction probability for the corresponding ground-truth class, resulting in data masks $m_1$ and $m_2$ denoting models $f_1$ and $f_2$ as teachers respectively:

$$m_1 = \mathbb{I}\left[\sigma(\mathbf{z}_1)_{gt} > \sigma(\mathbf{z}_2)_{gt}\right], m_2 = \mathbb{I}\left[\sigma(\mathbf{z}_1)_{gt} \leq \sigma(\mathbf{z}_2)_{gt}\right] \tag{3}$$

Here, $\sigma(\cdot)_{gt}$ denotes the softmax probability for the ground-truth class, and $\mathbb{I}$ represents the indicator function. This approach partitions the data into samples where the best performing model provides supervision to others while considering continuously improving models. The distillation loss is given as:

$$\mathcal{L}_{\mathrm{dist}} = m_1 \cdot \mathcal{L}_{KL_{2,1}} + m_2 \cdot \mathcal{L}_{KL_{1,2}} \tag{4}$$

Overall, the model exhibiting higher confidence acts as a teacher and the other as the student. The distillation loss is propagated only to the student model, and a stop-gradient operation is applied for the teacher. We also employ a task-specific loss (e.g., cross-entropy for image classification, $\mathcal{L}_T$ for every model) alongside the distillation loss to help preserve prior knowledge, and stabilize training. This combination constitutes the overall loss function:

$$\mathcal{L} = \mathcal{L}_{T_1} + \mathcal{L}_{T_2} + \mathcal{L}_{\mathrm{dist}} \tag{5}$$

### 3.3 Knowledge Transfer for dense tasks

We further extend our approach for semantic segmentation and video saliency prediction. Instead of the aforementioned confidence-based data partition, we assign the teacher role to the model having the lowest loss for each sample. Let $\mathcal{L}_T$ be the task-specific loss, the data masks are then formulated as:

$$m_1 = \mathbb{I}\left[\mathcal{L}_{T_1} < \mathcal{L}_{T_2}\right], \quad m_2 = \mathbb{I}\left[\mathcal{L}_{T_1} \geq \mathcal{L}_{T_2}\right] \tag{6}$$

these masks are utilized in Equation 4 to obtain the distillation loss $\mathcal{L}_{\mathrm{dist}}$, and the overall loss function is the same as in Equation 5.

**Semantic Segmentation:** For experiments on semantic segmentation, we use binary cross-entropy loss and dice loss (Milletari et al., 2016) for our mask loss:

$$\mathcal{L}_{\text{mask}} = \lambda_{\text{ce}}\mathcal{L}_{\text{ce}} + \lambda_{\text{dice}}\mathcal{L}_{\text{dice}} \tag{7}$$

where $\lambda_{\text{ce}} = 5.0$ and $\lambda_{\text{dice}} = 5.0$. The final task-specific loss for semantic segmentation is a combination of mask loss and classification loss:

$$\mathcal{L}_{\text{semseg}} = \mathcal{L}_{\text{mask}} + \lambda_{\text{cls}}\mathcal{L}_{\text{cls}} \tag{8}$$

with $\lambda_{\text{cls}} = 2.0$ for correct predictions and 0.1 for incorrect ones.

**Saliency Prediction:** Video saliency prediction utilizes a combination of Equation 1 and Correlation Coefficient (CC), which calculates the Pearson correlation between the ground-truth and the predicted saliency maps, as the final task-specific loss:

$$\mathcal{L}_{\text{VSP}} = \mathcal{L}_{\text{KL}} - \mathcal{L}_{\text{CC}} \tag{9}$$

$$\mathcal{L}_{\text{CC}} = \frac{\sigma(P, Q)}{\sigma(P, P) \times \sigma(Q, Q)} \tag{10}$$

here $P$ & $Q$ are the predicted saliency map and ground-truth respectively, and $\sigma(P, Q)$ represents the covariance between $P$ and $Q$.

### 3.4 Knowledge Transfer between multiple models

With the basic knowledge transfer between two models set up, extending it to parallel knowledge transfer between multiple models is the logical next step. As illustrated in Figure 2, the model with the highest prediction probability corresponding to the ground-truth class is selected as the teacher for a particular sample. Given $K$ models $f_0, f_1, \ldots, f_{K-1}$ parameterized by $\theta_0, \theta_1, \ldots, \theta_{K-1}$ respectively. The data mask for a model $k$ is computed as:

$$m_k = \mathbb{I}\left[\arg\max_k \left([\sigma(\mathbf{z}_0)_{gt}, \ldots, \sigma(\mathbf{z}_{K-1})_{gt}]\right) == k\right] \tag{11}$$

where $m_k$, with $k \in \{0, 1, \ldots, K-1\}$, represents whether the model $f_k$ acts as a teacher for a particular sample, and $[\sigma(\mathbf{z}_0)_{gt}, \ldots, \sigma(\mathbf{z}_{K-1})_{gt}]$ denotes the concatenation of prediction probabilities corresponding to the ground-truth class for $K$ models. The distillation loss is formulated as:

$$\mathcal{L}_{\text{dist}} = \sum_{i=0}^{K-1} \sum_{j=0, j\neq i}^{K-1} m_i \cdot \mathcal{L}_{KL_{j,i}} \tag{12}$$

with $\mathcal{L}_{KL_{j,i}}$ considering models $f_j$ and $f_i$ as student and teacher respectively. The overall loss is given as:

$$\mathcal{L} = \sum_{i=0}^{K-1} \mathcal{L}_{T_i} + \mathcal{L}_{\text{dist}} \tag{13}$$

# 4    Experiments & Results

In this section, we perform a series of experiments to evaluate our proposed method on image classification, semantic segmentation, and video saliency prediction benchmarks. Section 4.1 introduces the datasets used for various experiments, and Section 4.2 explains the followed training choices. Finally, we discuss our results on various tasks in Section 4.3.

## 4.1    Datasets

We verify the effectiveness of our approach on multiple tasks using the following datasets:

**ImageNet** (Deng et al., 2009) consists of 1.2 million images for training and 50,000 images for validation. We report the results of our knowledge transfer between two or multiple models on the validation set.

**ADE20K** (Zhou et al., 2017) provides 150 object and stuff categories, with 20,210 images in the training set and 2,000 images in the validation set. We use the validation set to evaluate our approach for knowledge transfer on semantic segmentation.

**DHF1K** (Wang et al., 2018) is a benchmark dataset for video saliency prediction, comprising 600 videos in the training set and 100 videos in the validation set. We use the validation set for our evaluation.

**Hollywood-2** (Mathe & Sminchisescu, 2014) is the largest dataset for video saliency prediction in terms of the number of videos, containing 1,707 clips sourced from 69 Hollywood movies. Following the standard evaluation protocol, we use the predefined split of 823 videos for training and the remaining 884 videos for testing.

## 4.2    Implementation details

We implement all the networks and training procedures in Pytorch (Paszke et al., 2019), and conduct all experiments on a single NVIDIA RTX A6000. We use the Adam optimizer for image classification and video saliency prediction, while experiments on semantic segmentation utilize the AdamW optimizer. For all experiments, the learning rate and weight decay are set to $1e$-6 and $1e$-5 respectively, with the temperature parameter set to 1 in Equation 1. All models are trained in full precision for 20 epochs with a batch size of 128. We do not apply any data augmentations, learning rate schedulers, or layer-wise learning rate decay. The only exceptions are ViTs, for which we employ the default data augmentations and cosine scheduler provided by the *timm* (Wightman et al., 2019) library.

For experiments on ImageNet, we compare our approach with Roth et al. (2024). We independently execute the KL+DP framework twice, each time initializing with the original models. The role of student and teacher is swapped in the second run. The original KL+DP paper presents experiments involving cross-entropy loss, which were shown to degrade performance. We observed a similar decline during our replication of their method. Therefore, to ensure a fair comparison , we adopt their best performing configuration, excluding cross-entropy losses. Additionally, we compare against the supervised variant of KL+DP, wherein data partitioning is guided by the model's confidence with respect to the ground-truth labels.

For knowledge transfer between semantic segmentation models, we utilize Mask2Former (Cheng et al., 2022) and follow its original experimental setup. Finally, we perform knowledge transfer between state-of-the-art video saliency models proposed by Zhou et al. (2023) and Girmaji et al. (2025). We adopt their respective experimental setups to train the models from scratch, before transferring knowledge between them.

## 4.3    Results

**Image Classification.**    For evaluating our approach on ImageNet, we utilize pretrained models listed in Table 1. The models were chosen to cover a wide range of architectures, performance levels, sizes, and training objectives. We utilize the base and small variants of MAE and DINOv2, respectively. In Table 2, we report $\Delta_{\text{top-1}}$, which represents the change in Top-1 accuracy after transferring knowledge between models. Since Roth et al. (2024) have demonstrated that standard KD can negatively impact performance when

Table 1: Selection of models used for experiments on the validation set of ImageNet.

| Models | Acc. | # Params. (M) |
|---|---|---|
| SeNet154 (He et al., 2019) | 81.378 | 115.09 |
| SWSL-ResNext101 (Xie et al., 2017) | 84.276 | 88.79 |
| MAE (He et al., 2022) | 83.446 | 86.57 |
| ViT-B (Dosovitskiy et al., 2021) | 79.152 | 86.57 |
| PiT-B (Heo et al., 2021) | 82.278 | 73.76 |
| ResMLP-36 (Touvron et al., 2022) | 79.576 | 44.69 |
| MambaVision-T2 (Hatamizadeh & Kautz, 2024) | 82.506 | 35.1 |
| ResMLP-24-dist (Touvron et al., 2022) | 80.548 | 30.02 |
| DINOv2 (Oquab et al., 2023) | 81.332 | 23.98 |
| ViT-S (Dosovitskiy et al., 2021) | 78.842 | 22.05 |
| CoaT-lite-mini (Xu et al., 2021) | 78.858 | 11.01 |
| PiT-XS (Heo et al., 2021) | 77.916 | 10.62 |
| ViT-T (Dosovitskiy et al., 2021) | 75.466 | 5.72 |

Table 2: Comparative results of change in Top-1 accuracy after transferring knowledge between models pretrained on ImageNet using different methods. We conduct three runs for our method and report the mean and standard deviations.

| Method | Model 1 | $\Delta_{top\text{-}1}$ | Model 2 | $\Delta_{top\text{-}1}$ |
|---|---|---|---|---|
| KL+DP
Ours | DINOv2 | 0.582
**1.494 ± 0.042** | MAE | 0.306
**0.426 ± 0.028** |
| KL+DP
Ours | PiT-B | 0.73
**0.834 ± 0.029** | SWSL-ResNext101 | **0.336**
0.228 ± 0.012 |
| KL+DP
Ours | DINOv2 | 0.968
**1.46 ± 0.042** | MambaVision-T2 | -0.242
**0.061 ± 0.018** |
| KL+DP
Ours | DINOv2 | 0.818
**1.587 ± 0.012** | SWSL-ResNext101 | **0.538**
0.431 ± 0.036 |
| KL+DP
Ours | CoaT-lite-mini | 0.436
**0.515 ± 0.03** | SeNet154 | **0.48**
0.477 ± 0.017 |
| KL+DP
Ours | CoaT-lite-mini | 0.386
**0.567 ± 0.026** | DINOv2 | 0.692
**1.381 ± 0.051** |
| KL+DP
Ours | PiT-XS | 0.37
**0.477 ± 0.01** | ResMLP-36 | 0.086
**0.253 ± 0.021** |
| KL+DP
Ours | CoaT-lite-mini | 0.17
**0.425 ± 0.032** | PiT-XS | 0.222
**0.495 ± 0.033** |
| KL+DP
Ours | ViT-B | 0.614
**1.163 ± 0.009** | ViT-S | 0.518
**0.678 ± 0.007** |
| KL+DP
Ours | ViT-B | 0.492
**1.385 ± 0.005** | ViT-T | 0.828
**0.895 ± 0.003** |
| KL+DP
Ours | ViT-S | 0.336
**0.833 ± 0.003** | ViT-T | 0.724
**0.949 ± 0.002** |

learning from weaker or similarly performing teacher models, we compare our proposed method with their approach, referred to as KL+DP.

From the results presented in Table 2, we observe that Bi-KD consistently improves the performance of both participating models. This finding substantiates our hypothesis that pretrained models serve as effective sources for transferring complementary knowledge between one another, thereby enabling the enhancement of each model independently. Furthermore, our method demonstrates superior performance compared to KL+DP, outperforming it in 19 out of 22 cases. The results provide supporting evidence for our hypothesis, demonstrating that simultaneous, bidirectional knowledge transfer, enabled through Bi-KD is more effective than the unidirectional knowledge transfer employed by KL+DP, where the frozen teacher model is not optimized for teaching.

Our experiments further indicate that the most significant performance improvements occur when models with differing training methodologies are paired together. For example, pairing the self-supervised DI-

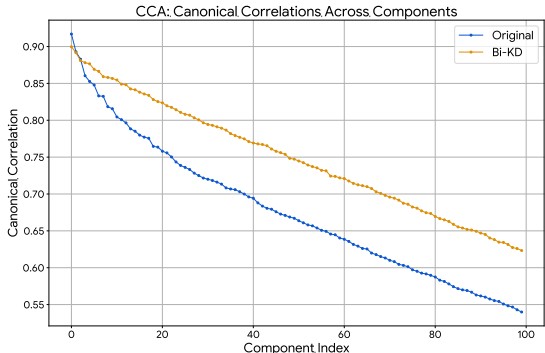

Figure 3: Canonical Correlation Analysis (CCA) of the Top-100 canonical components between the feature representations of ViT-S and ViT-T. The plot compares correlations from the original models with those from Bi-KD, showing that Bi-KD achieves stronger feature-space alignment.

Table 3: Comparison of each model pair's performance before and after knowledge transfer (first training run) using Bi-KD, along with the performance of their direct ensemble.

| Model 1 | Top-1 | | Model 2 | Top-1 | | Ensemble | Recovered (%) | |
|---|---|---|---|---|---|---|---|---|
| | Before | After | | Before | After | | Model 1 | Model 2 |
| DINOv2 | 81.332 | 82.722 | MAE | 83.446 | 83.842 | 84.332 | 46.3 | 44.7 |
| PiT-B | 82.278 | 83.1 | SWSL-ResNext101 | 84.276 | 84.506 | 85.206 | 28.1 | 24.7 |
| DINOv2 | 81.332 | 82.804 | Mamba Vision-T2 | 82.506 | 82.542 | 83.56 | 66 | 3.4 |
| DINOv2 | 81.332 | 82.902 | SWSL-ResNext101 | 84.276 | 84.75 | 85.162 | 40.1 | 53.5 |
| CoaT-lite-mini | 78.858 | 79.34 | SeNet154 | 81.378 | 81.834 | 82.302 | 14 | 49.4 |
| CoaT-lite-mini | 78.858 | 79.442 | DINOv2 | 81.332 | 82.694 | 82.732 | 15 | 97.3 |
| PiT-XS | 77.916 | 78.388 | ResMLP-36 | 79.576 | 79.85 | 80.242 | 20.3 | 41.1 |
| CoaT-lite-mini | 78.858 | 79.238 | PiT-XS | 77.916 | 78.366 | 79.866 | 37.7 | 23.1 |
| ViT-B | 79.152 | 80.326 | ViT-S | 78.842 | 79.526 | 80.48 | 88.4 | 41.8 |
| ViT-B | 79.152 | 80.544 | ViT-T | 75.466 | 76.364 | 80.274 | 124 | 18.7 |
| ViT-S | 78.842 | 79.68 | ViT-T | 75.466 | 76.412 | 79.69 | 98.8 | 22.4 |

NOv2 (Oquab et al., 2023) with any model trained using a supervised objective consistently results in performance improvements exceeding 1%. Interestingly, SWSL-ResNext101 (Xie et al., 2017), which has a Top-1 accuracy of 84.276%, benefits more when paired with the relatively weaker DINOv2 than with the stronger PiT-B (Heo et al., 2021). Similarly, CoaT-lite-mini (Xu et al., 2021) shows a greater improvement when paired with DINOv2 than with SeNet154 (He et al., 2019), despite both having comparable performance. These results align with the observation made by Gontijo-Lopes et al. (2022) that models trained through different methodologies tend to make uncorrelated errors, thereby making even lower performing models valuable contributors in knowledge transfer.

Another notable observation is that knowledge transfer between models with the same architecture also results in considerable performance improvements. All three models: ViT-B, ViT-S, and ViT-T, consistently yield performance improvements when paired with each other. Interestingly, both ViT-B and ViT-S experience greater performance gains when paired with the smaller ViT-T, rather than with each other. This

further underscores that even smaller models can capture data-specific insights that may be absent in larger counterparts.

**Distillation vs Ground-Truth Labels:** We also examine how distillation labels deviate from ground-truth labels. Since the teacher model is chosen based on having higher confidence in the ground-truth class, three distinct scenarios arise:

- **Case 1** - Both the teacher and the student correctly predict the ground-truth class.

- **Case 2** - The teacher predicts the correct class, but the student predicts incorrectly.

- **Case 3** - Both the teacher and the student predict incorrectly, but the teacher assigns higher confidence to the ground-truth class than the student.

In Cases 1 and 2, the student clearly benefits from distillation. Although Case 3 may negatively impact training, it is worth noting that the teacher still demonstrates better alignment with the ground-truth. Additionally, the task-specific loss function contributes to training stability and mitigates the potential adverse effects of Case 3.

Analyzing the predictions of original ViT-S and ViT-T models on the validation set, we find that Case 1 accounts for 70.9%, Case 2 for 12.5% and Case 3 for 16.6% of cases. Post Bi-KD, Case 1 increases to 73.1%, Case 2 decreases to 10.1%, while Case 3 remains at similar levels, suggesting limited knowledge transfer in Case 3. Overall, the above empirical evidence indicates predominant benefits of using Bi-KD (Case 1 and Case 2) and minimal adverse effects when distillation differs from ground truth labels (Case 3).

**Feature Evolution:** We further inquire into the evolution of feature representations across the participating models, specifically, whether they become more alike following Bi-KD. To this end, we employ Canonical Correlation Analysis (CCA) to quantify and compare feature representations of the two models, before and after Bi-KD. The corresponding plot for ViT-S and ViT-T models is shown in Figure 3. CCA results show consistently higher correlations after Bi-KD, indicating that feature representations converge more closely between models. Considering the Top-100 components, we compute the mean CCA over the CLS embeddings from the layer preceding the MLP head, and observe an improvement from 0.6768 to 0.7487 post Bi-KD. This suggests that Bi-KD not only improves accuracy but also fundamentally aligns and enhances the models' internal representations.

Finally, Table 3 presents a comparison between the performance of Bi-KD and a direct ensemble of its two constituent models. For the ensemble, the final classification is obtained by averaging the softmax scores of the individual models. To quantify this comparison, we calculate the percentage of the ensemble's performance retained by each model using the following formula:

$$Recovered = \frac{\text{Top-1}_{\text{after}} - \text{Top-1}_{\text{before}}}{\text{Ensemble} - \text{Top-1}_{\text{before}}} \tag{14}$$

As shown in Table 3, our method allows individual models to recover more than half of the ensemble's performance in most scenarios. Larger models, such as SeNet154, SWSL-ResNext101, MAE (He et al., 2022), ViT-B, and PiT-B, prove particularly effective, recovering approximately 60% of the ensemble performance on average. Among these, ViT-B stands out by recovering nearly 90% of the ensemble accuracy in one instance and surpassing the ensemble performance in another.

In contrast, smaller models, such as DINOv2, ViT-S, CoaT-lite-mini, PiT-XS (Heo et al., 2021), and ViT-T, recover roughly 40% of the ensemble's performance on average. In particular, DINOv2 and ViT-S perform consistently well, each recovering at least 40% of the ensemble performance in all cases and closely approaching it in some instances.

Overall, these findings underscore the effectiveness of our approach in enabling individual models, both large and small, to recover a noticeable portion of the ensemble-level performance.

Table 4: Performance comparison of Mask2Former models before and after knowledge transfer on ADE20K dataset. We conduct five runs for each experiment and report the mean and standard deviations.

| Backbones | Before | | | | After | | | |
|---|---|---|---|---|---|---|---|---|
| | mIoU | fwIoU | mACC | pACC | mIoU | fwIoU | mACC | pACC |
| R50 | 47.23 | 70.95 | 60.11 | 81.72 | $47.78 \pm 0.12$ | $71.15 \pm 0.07$ | $60.47 \pm 0.13$ | $81.88 \pm 0.07$ |
| Swin-T | 47.7 | 72.37 | 61.44 | 82.7 | $48.32 \pm 0.07$ | $72.54 \pm 0.07$ | $61.8 \pm 0.04$ | $82.87 \pm 0.07$ |
| Swin-S | 51.33 | 73.26 | 65.11 | 83.48 | $51.77 \pm 0.15$ | $73.44 \pm 0.06$ | $65.44 \pm 0.14$ | $83.6 \pm 0.05$ |
| Swin-T | 47.7 | 72.37 | 61.44 | 82.7 | $48.45 \pm 0.13$ | $72.51 \pm 0.05$ | $62.08 \pm 0.2$ | $82.84 \pm 0.05$ |

Table 5: We apply Bi-KD between TMFI-Net and ViNet-A, and compare the resulting models against a range of individually trained saliency prediction methods. The last two rows represent the Bi-KD variants.

| Models | DHF1K | | Hollywood-2 | |
|---|---|---|---|---|
| | CC | NSS | CC | NSS |
| ViNet (Jain et al., 2021) | 0.521 | 2.957 | 0.693 | 3.73 |
| TSFP-Net (Chang & Zhu, 2021) | 0.529 | 3.009 | 0.711 | 3.91 |
| STSA-Net (Wang et al., 2021) | 0.539 | 3.082 | 0.705 | 3.908 |
| TMFI-Net (Zhou et al., 2023) | 0.552 | 3.188 | 0.737 | 4.054 |
| THTD-Net (Moradi et al., 2024) | 0.553 | 3.188 | 0.726 | 3.965 |
| ViNet-S (Girmaji et al., 2025) | 0.529 | 3.008 | 0.728 | 3.941 |
| ViNet-A (Girmaji et al., 2025) | 0.525 | 3.019 | 0.756 | 4.119 |
| TMFI-Net (Zhou et al., 2023) (Bi-KD) | **0.558** | **3.216** | 0.75 | 4.148 |
| ViNet-A (Girmaji et al., 2025) (Bi-KD) | 0.536 | 3.077 | **0.762** | **4.198** |

**Semantic Segmentation.** Table 4 compares the performance of Mask2Former models with various backbones before and after applying our proposed knowledge transfer method. The models are evaluated using four standard semantic segmentation metrics: mean Intersection-over-union (mIoU), frequency weighted Intersection-over-union (fwIoU), mean Accuracy (mACC), and pixel Accuracy (pACC).

Our approach leads to consistent improvements in each metric within a single stage of training, whether transferring knowledge between different architectures or similar ones. Notably, Swin-T (Liu et al., 2021) backbone sees its mIoU increase from 47.7 to 48.32 when paired with similarly performing R50 (He et al., 2016) backbone, which further increases to 48.45 when paired with the stronger Swin-S (Liu et al., 2021) backbone. While the improvements are not dramatic, their consistency demonstrates the utility of our proposed approach in extracting additional performance from already well-trained models.

**Video Saliency Prediction.** Table 5 presents a comparative evaluation of video saliency prediction task on the DHF1K and Hollywood-2 datasets using two standard metrics: CC and Normalized Scanpath Saliency (NSS). We perform knowledge transfer on the TMFI-Net (Zhou et al., 2023) and ViNet-A (Girmaji et al., 2025) models. The performance of their Bi-KD variants is compared against their original versions as well as other state-of-the-art methods.

On the DHF1K dataset, TMFI-Net (Bi-KD) establishes a new state-of-the-art, improving its CC from 0.552 to 0.558 and its NSS from 3.188 to 3.216. Similarly, ViNet-A (Bi-KD) demonstrates consistent gains, with its CC increasing from 0.525 to 0.536 and NSS from 3.019 to 3.077.

On the Hollywood-2 dataset, the performance gains are even more substantial. TMFI-Net (Bi-KD) improves from a CC of 0.737 to 0.750 and from an NSS of 4.054 to 4.148. ViNet-A (Bi-KD) achieves a new state-of-the-art, raising its CC from 0.756 to 0.762 and its NSS from 4.119 to 4.198.

These results validate the efficacy of our approach, demonstrating that mutual knowledge transfer consistently enhances performance across diverse architectures and datasets, thereby advancing the state-of-the-art in video saliency prediction.

Table 6: Parallel multidirectional knowledge transfer across multiple models. For image classification, we report results for two-way, three-way, and four-way knowledge transfer using our proposed approach. For video saliency prediction, we present results for two-way and three-way knowledge transfer.

| Models | $\Delta_{\text{top-1}}$ |
|---|---|
| CoaT-lite-mini | 0.38 |
| PiT-XS | 0.45 |
| CoaT-lite-mini | 0.46 |
| PiT-XS | 0.476 |
| ResMLP-24-dist | 0.15 |
| CoaT-lite-mini | 0.572 |
| PiT-XS | 0.63 |
| ResMLP-24-dist | 0.286 |
| DINOv2 | 1.28 |

(a) Image classification

| Models | DHF1K | |
|---|---|---|
| | CC | NSS |
| TMFI-Net | 0.558 | 3.216 |
| ViNet-A | 0.536 | 3.077 |
| TMFI-Net | **0.561** | **3.224** |
| ViNet-A | 0.54 | 3.087 |
| ViNet-S | 0.533 | 3.038 |

(b) Video saliency prediction

**Multi-directional Transfer.** Finally, we extend our knowledge transfer framework to support parallel, multidirectional transfer among multiple models within a single training stage. Table 6 demonstrates the effectiveness of our parallel multidirectional knowledge transfer strategy across both image classification and video saliency prediction tasks.

In image classification on ImageNet, all participating models consistently benefit as more models are incorporated into the collaborative learning setup. For example, the performance gain for CoaT-lite-mini increases from 0.38 to 0.46, and PiT-XS improves from 0.45 to 0.476 when ResMLP-24-dist (Touvron et al., 2022) is added. These gains are further amplified, reaching 0.572 for CoaT-lite-mini, 0.63 for PiT-XS, and 0.286 for ResMLP-24-dist, with the inclusion of DINOv2. These results highlight the scalability and effectiveness of our approach with respect to the number of participating models.

Importantly, our approach is also task-agnostic. In video saliency prediction on DHF1K, TMFI-Net benefits from the knowledge transferred from ViNet-A, achieving a CC of 0.558 and an NSS of 3.216. Incorporating ViNet-S (Girmaji et al., 2025) into the collaborative learning environment further boosts the performance of both TMFI-Net and ViNet-A, with TMFI-Net achieving a new state-of-the-art. These findings affirm the robustness and generality of our knowledge transfer strategy across both architectures and tasks.

## 5 Conclusion

In this work, we introduce a simple yet effective approach for simultaneous multidirectional knowledge transfer between pretrained models. Our method employs a dynamic data partitioning scheme that selects the most suitable teacher model for each sample, resulting in consistent performance improvements across all participating models within a single training stage. By enabling each model to serve as both a learner and a teacher, our framework fosters mutual enhancement and contributes to the improvement of participating models. We demonstrate the effectiveness of our approach across a range of model architectures and tasks, including image classification, semantic segmentation, and video saliency prediction. Notably, our method sets a new state-of-the-art in video saliency prediction, underscoring the potential of collaborative knowledge transfer in complex visual understanding tasks. Additionally, we extend our framework to support knowledge transfer among multiple models and find that performance continues to improve as more models are added to the collaborative environment. Our results provide compelling evidence for the viability of simultaneous multidirectional knowledge transfer between pretrained models.

Our approach has certain limitations. It builds on models pretrained for a specific task and transfers knowledge between them, which may affect the generalization capabilities of the underlying models. This concern is more pronounced for architectures like DINOv2, whose backbones are widely adopted across diverse tasks. Moreover, the method depends on ground-truth labels, as a task-specific loss is employed to stabilize training, restricting its use in fully unsupervised settings. Addressing these limitations will be an

important direction for future work, which may include exploring model merging to consolidate the strengths of multiple models into a single, higher-performing system.

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
