# OpenReview forum: "Simplifying Knowledge Transfer in Pretrained Models"
_TMLR — Accepted by TMLR_

### Review · Reviewer_3Rzi · 2025-07-11

**Summary Of Contributions:**

The paper presemnts a simplet yet effective strategy for knowledge transfer via model distillation. The proposed approach leverages multiple pre-trained models, and each one of them can serve as the teacher for training samples, based on some loss metric. The experiments evaluated this approach on various tasks and pre-trained model pairs, showing improved performance over existing methods.

**Audience:**

Yes

**Broader Impact Concerns:**

N.A

**Claims And Evidence:**

No

**Requested Changes:**

See weaknesses above.

**Strengths And Weaknesses:**

**Strength**
1. The paper is clearly written.
2. The experiments are extensive with various pre-trained models and tasks.

***Weaknesses*
1. Some crucial technical details are missing. For the distillation loss, there appears to be no stop-gradient operation. This may be problematic as it allows both models trying to match the prababilities of the other. Could the authors clarify this point?
2. Each model is learning simultaneously from both the ground truth labels and distillation labels, why is this necessary? The current manuscript does not explain this design choice.
3. In a related question, how much do ground truth labels differ from the distillation labels? What if distillation labels actually predict a wrong ground truth class despiting being the more confident prediction among the two models? In other words, there needs to be some justifications/analysis for teacher selection based on loss metric.
4. Since the paper is mostly empirical, I would expect more interprretation results explaining why the method works well in practice, instead of just showing the end results that it outperforms the baseline. For instance, does the performance improvements come from the teachers being dynamic instead of being fixed in KL + DP? Does KL + DP also use ground truth labels as part of the loss (in other words is the comparison fair)?
5. The comparison focuses primarily on the single baseline of KL + DP.  This may be insufficient to justify the efficacy of the proposed approach (alternative baselines such as self-distllation etc).
6. Some of the improvements are marginal (Table 4) and reported with any confidence level. This makes it hard to assess the method's efficacy. In addition, all performance are worse than the ensemble (Table 3), this may suggest some sub-optimal teacher selection for the distillation targets. Also it raises the question of why not using the ensemble probabilities as distiallation target?

---

> ### Author Response · Authors · 2025-07-29
> **Response to Review (1/2)**
>
> We thank the reviewer for their valuable and positive feedback. We address their concerns below:
>
> **Stop-gradient:** A stop-gradient operation is applied for the teacher. The model with higher confidence serves as the teacher, while the distillation loss is propagated only to the less confident model. Specifically, when Model 1 exhibits higher confidence, it acts as the teacher and only Model 2 receives the distillation loss, and vice versa. We will clarify this in the final version of the paper.
>
> **Ground truth labels and distillation:** In the absence of task-specific losses (e.g. cross-entropy loss on ground truth labels for classification task), our method demonstrates unstable training, likely attributable to inadequate preservation of prior knowledge. We will clarify this in the final version of the paper.
>
> **KL+DP with ground truth labels:** Our experiments found that applying cross entropy loss to KL+DP leads to performance degradation (the same is also discussed in their paper).  To ensure a fair comparison, we adopt their best-performing configuration, which excludes cross-entropy losses. However, it must be noted that we compare against the supervised variant of KL+DP, where data partitioning is based on model confidence relative to ground-truth labels.
>
> **Marginal improvements:** While the improvements are marginal in a few cases (e.g. Table 4), they are consistent across tasks and models, which is of notable significance. Moreover, our method achieves state-of-the-art performance on the saliency prediction task, giving significant gains.
>
> **Distillation with ensembles**: Thank you for the valuable suggestion. As suggested, we tried an approach of distillation with ensembles. We find that in the classification task, it gives competent results when compared to Bi-KD (results on some backbones shown in Table 1 below). However, it fails to match the performance on the saliency prediction task and leads to slight decrease in the NSS metric (Table 2 below). We will add these experiments for all backbones in the final version of the paper.
>
> **Table 1**
> | Method   | Model 1      | $\Delta_{top-1}$ | Model 2         | $\Delta_{top-1}$ |
> |----------|--------------|------------------|-----------------|------------------|
> | Ensemble    | DINOv2       | 1.54            | SWSL-ResNext101             | 0.384            |
> | **Ours** |              | **1.57**         |                 | **0.474**        |
> | Ensemble    | CoaT-lite-mini        | **0.616**             | DINOv2 | 1.342            |
> | **Ours** |              | 0.584        |                 | **1.364**             |
> | Ensemble    | CoaT-lite-mini       | **0.402**            | PiT-XS  | **0.482**           |
> | **Ours** |              | 0.38        |                 | 0.45        |
> | Ensemble    | ViT-B       | **1.418**            | ViT-T  | 0.866           |
> | **Ours** |              | 1.392        |                 | **0.898**        |
> | Ensemble    | ViT-S       | 0.83            | ViT-T  | **0.946**           |
> | **Ours** |              | **0.838**        |                 | **0.946**        |
>
> **Table 2**
> | Method    | Model     | CC    | NSS   |
> |-----------|-------------|-------------|-------------|
> | Ensemble  | TMFI‑Net    | 0.553       | 3.185       |
> |                   | ViNet-A       | 0.533       | 3.055       |
> | **Ours**    | TMFI-Net    | **0.558**  | **3.216**  |
> |                  | ViNet-A       | **0.536**  | **3.077**  |
>
> **Self-Distillation:** In self-distillation, the student model is trained from scratch, whereas Bi-KD begins with two pre-trained models. As a result, self-distillation requires significantly more computational resources and hyperparameter tuning. Nevertheless, in response to the reviewer’s suggestion, we applied self-distillation using the ViT-S architecture. Notably, self-distillation was run for over 300 epochs, while Bi-KD required only 20 epochs of fine-tuning.
>
> Specifically, we conducted self-distillation using a pre-trained ViT-S teacher to train an uninitialized ViT-S student model. The self-distillation setup achieved 79.568% accuracy, compared to 79.68% from Bi-KD using ViT-T as the teacher.

---

> > ### Author Response · Authors · 2025-07-29
> > **Response to Review (2/2)**
> >
> > **How much they differ from ground labels**: The model exhibiting higher confidence in the ground-truth label is designated as the teacher.
> > This gives rise to three possible scenarios:
> >
> > * Case 1: Both the teacher and the student predict the correct class.
> > * Case 2: The teacher predicts the correct class, while the student predicts incorrectly.
> > * Case 3: Both the teacher and the student predict incorrectly, but the teacher assigns higher confidence to the ground-truth class than the student.
> >
> > At the beginning of training with ViT-S and ViT-T models, Case 1 occurs in 70.8% of instances, Case 2 in 12.3%, and Case 3 in 16.5%. These proportions represent the initial distribution and evolve over time.
> >
> > In Cases 1 and 2, the student clearly benefits from distillation. Although Case 3 may negatively impact training, it is worth noting that the teacher still demonstrates better alignment with the ground truth. Additionally, the task-specific loss function contributes to training stability and mitigates the potential adverse effects of Case 3.

---

> ### Author Response · Authors · 2025-08-11
> **Response to Review**
>
> Dear reviewer 3Rzi, thank you for your time in going over our response. We would be happy to provide additional clarification or details if necessary over the last week of the discussion phase.

---

> ### Author Response · Authors · 2025-08-16
> **Response to Review**
>
> Dear reviewer 3Rzi, the manuscript has been revised accordingly, and we remain available to provide any further clarifications or to address additional feedback as needed.

---

### Review · Reviewer_ppWT · 2025-07-22

**Summary Of Contributions:**

This paper builds on a previous method [1] for data partitioning to achieve the transfer of complementary knowledge between pretrained models. Unlike the previous method, which has a frozen teacher and trainable student, the paper proposes to dynamically switch the roles of the models, thus transferring the complementary knowledge bidirectionally. The proposed method is also easily transferrable to groups of multiple models.

**Audience:**

Yes

**Broader Impact Concerns:**

I have no such concerns.

**Claims And Evidence:**

Yes

**Requested Changes:**

See above.

**Strengths And Weaknesses:**

**Strengths:**
1. The proposed method is straightforward and shows competitive results.
2. The proposed method extends to more than the classification task, including dense prediction and video saliency prediction, which were not discussed in previous papers.

**Weaknesses:**
1. The experiment details are not comprehensive. How the baseline results are derived seems ambiguous. See the Questions.
2. Roth et al. [1] shows that in an unsupervised setting, in which they partition the data based on model confidence and do not use the classification loss, they are able to achieve similar results using a classification-guided loss. The authors do not discuss whether a similar unsupervised setting is effective for their method.

**Questions:**
1. For the baseline method, it is unclear how the results are derived. Specifically, for the student model (i.e., model 1), it is stated to be distilled from the teacher model (i.e., model 2). However, it remains ambiguous whether the teacher model is distilled from the original student model or from the already distilled student model. Additionally, it is not specified whether the loss function used in the baseline includes the cross-entropy loss alongside the distillation loss. Clarification on these points would help better interpret the reported performance.
2. Roth et al. [1] partition the training data into two subsets, using one subset for distillation from a frozen copy of the student model to retain previously learned knowledge. In contrast, the proposed method lacks such an explicit mechanism for preserving prior knowledge. Notably, by skipping training on non-complementary data, resulting in fewer update iterations, the method still achieves improved performance. Could the authors clarify whether distilling from a frozen student model is necessary or beneficial in this context?
3. The proposed method uses a bidirectional distillation loss. When computing gradients with respect to the parameters of the first model, both directions of the loss contribute to the update. On complementary data, the loss encourages the student model to align with the teacher, while on non-complementary data, it promotes divergence from the teacher. Could this dynamic of combining alignment and contrast depending on the data contribute to the improved performance?

[1] Roth, K., Thede, L., Koepke, A.S., Vinyals, O., Hénaff, O. and Akata, Z., 2023. Fantastic gains and where to find them: On the existence and prospect of general knowledge transfer between any pretrained model. arXiv preprint arXiv:2310.17653.

---

> ### Author Response · Authors · 2025-07-29
> **Response to Review**
>
> We thank the reviewer for their positive and valuable feedback. We address their concerns below:
>
> **Clarity on baseline method:** We independently execute the KL+DP framework twice, each time initializing with the original models. The role of student and teacher is swapped in the second run. The original KL+DP paper presents experiments involving cross-entropy loss, which were shown to degrade performance. We observed a similar decline during our replication of their method. Therefore, to ensure a fair comparison, we adopt their best-performing configuration, excluding cross-entropy losses. Additionally, we compare against the supervised variant of KL+DP, wherein data partitioning is guided by the model’s confidence with respect to the ground-truth labels.
>
>
> **Data partitioning:** Initially, our method results in similar data partitions, one where the student is better and one where the teacher is better. The key difference is that both models learn from each other, allowing the partition to evolve over time. Additionally, we employ a task-specific loss (e.g., cross-entropy) to help preserve prior knowledge. Our approach is particularly advantageous in multi-model settings, as all n models are trained simultaneously, unlike KL+DP which requires n separate iterations, each with a different student.
>
> **Bidirectional loss:** A stop-gradient operation is applied during distillation on the non-complementary data. The model with higher confidence serves as the teacher, while the distillation loss is propagated only to the less confident model. Specifically, when Model 1 exhibits higher confidence, it acts as the teacher and only Model 2 receives the distillation loss, and vice versa. We will clarify this in the final version of the paper.
>
> **Unsupervised setting:** Without task-specific losses, our method exhibits unstable training, likely due to insufficient retention of prior knowledge. We will clarify and acknowledge this as a limitation in the final version of the paper.

---

> ### Author Response · Authors · 2025-08-11
> **Response to Review**
>
> Dear reviewer ppWT, thank you for your time in going over our response. We would be happy to provide additional clarification or details if necessary over the last week of the discussion phase.

---

> ### Author Response · Authors · 2025-08-16
> **Response to Review**
>
> Dear reviewer ppWT, the manuscript has been revised accordingly, and we remain available to provide any further clarifications or to address additional feedback as needed.

---

### Review · Reviewer_XJcU · 2025-07-24

**Summary Of Contributions:**

The paper presents an automated collaborative transfer-learning approach between 2 models via sample-wise peer knowledge distillation. In essence, the authors train two pretrained models jointly, and treat the one with the higher probability on the true label for a given sample as the teacher and its logits are distilled (KL, T=1) into the other. Roles (teacher and student) can swap from sample to sample based on the logits as a measure of confidence or certainty. In dense tasks, this is replaced by selecting the model with minimum loss as the teacher on the same per sample basis. *Data partitioning* is implemented as a binary mask that routes the KL loss from the currently more label-confident peer; all models still receive CE on all samples. The ensemble is then said to form an ensemble based "collaborative learning environment", where models jointly distill from one another.

The central contribution of the authors is the aforementioned methodology for mutual knowledge distillation. Implicitly, the narrative suggests that this constitutes a step toward an automated “self-improving zoo” with minimal changes, no extra teachers or temperatures, and applicability to heterogeneous architectures, however, experiments are limited to two models ($K=2$) trained in joint bi-distillation, although the paper outlines a multi-model approach (presumably $K > 2$). The authors report modest gains (+0.5–1.5% top‑1 on ImageNet for ViT-B paired with a smaller ViT-T, as well as improved segmentation on ADE20k video saliency), and claim to *recover* a large share of the ensemble’s boost inside a single model, argued via a *Recovered % of ensemble improvement* metric. However, it is unknown how much training were performed to achieve said gains, as this is unreported.

**Audience:**

Yes

**Broader Impact Concerns:**

The paper is a work on knowledge distillation, and has no applicable broader impact concerns from the viewpoint of this reviewer.

**Claims And Evidence:**

No

**Requested Changes:**

1. The paper should include some additional downstream tasks to show how the model generalises beyond the datasets they were trained on. Additionally, some OOD analysis should be performed, particularly with foundational models such as DINOv2, which has been trained for wide adoption to different downstream tasks / baselines. This is critical for acceptance.
2. The narrative of an automated joint collaborative training is interesting, but somewhat oversold. The paper presents results for bi-distillation, and hence the narrative should reflect this. Reformulating this so the contribution is clarified is necessary for acceptance.
3. An analysis of training dynamics, and a post-training analysis on the representation spaces would go a long way to help the reader evaluate the merits of the method. An analysis on how the training dynamics adjust to small differences and ties during gating between the two models is a good start. This reviewer asks themselves; "did the representations become more alike after mutual distillation"? This could be tested via a canonical-correlation analysis or mutual information estimate, although the former is simpler to perform. While this would improve the paper tremendously, it is not absolutely necessary for acceptance.
4. Without details on training and implementation, readers have no hope of being able to reproduce the reported results. At the bare minimum, batch size, number of training steps, and a general training setup should be included. This is crucial for acceptance.

**Strengths And Weaknesses:**

### Strengths

1. The authors should be commended for highlighting a highly ambitious and quite interesting research direction towards automated joint learning for model ensembles. While mutual distillation has been explored, the idea of extending this to automated ensembles is an interesting research direction that warrants further exploration.
2. The approach is demonstrated to generalise across different tasks, including classification, segmentation, and saliency. Consistent gains in results show promise. In particular, the model provides strong results in video saliency.
3. The approach requires little modification and additional machinery to provide results, and could potentially be used as a drop-in replacement for existing distillation and fine-tuning pipelines.

### Weaknesses

1.  A main concern of this reviewer is generalisation. The models are evaluated on a singular task, and exclusively with models that are pretrained for the given task. No hold-out sets or different downstream tasks are evaluated, which raises concerns on overfitting.
2. A second concern is how the core gating mechanism used in bi-distillation depends on the ground-truth label, which is a signal unavailable at test time or on unlabeled data. However, the narrative leans on a *self-improving zoo*. In contrast, an EMA distillation process is cheaper and can arguably be posed as a bi-distillation process that is amenable to SSL and importantly, does not require two hot models in memory simultaneously to perform training. This is known to be a computational bottleneck, particularly in terms of memory.
3. Hard, argmax-based masking is nondifferentiable and could be unstable around ties, and the training stability/variance isn’t reported. How does the distillation perform when differences are small or tied?
4. Uniform KL with $T=1$ between near-one-hot vectors closely mirrors cross entropy, which makes it unclear how much dark knowledge [1] is actually distilled between the models. Again, an analysis of the dynamics in training would be a welcome contribution, and would help readers gain better understanding of the method.
5. The paper is missing several training and implementation details. Batch sizes, epoch/step counts, augmentation pipelines, exact TIMM configs/checkpoint hashes, gradient clipping, AMP settings, training time / GPU-hours are not mentioned. While not strictly required, an anonymised GitHub repo with config files could potentially substitute for this omission, but as it stands, the reported results are not reproducible. In particular; details on specific model capacities for some models (e.g. DINOv2 and MAE) are missing. What was the model capacity (Base, or Large) for these models?

### Summary

The paper broaches an interesting topic on collaborative learning, and is ambitious in framing its narrative. However, the work seems to somewhat implicitly overpromise based on the abstract and introduction, limiting the experiments to bi-distillation. The work is lacking an analysis on how learning dynamics affect training. The paper is tersely written and sparse in detail, resulting in a work that is currently non-reproducible. This reviewer's impression is that the manuscript requires serious revision to be considered for acceptance.

[1] Distilling the Knowledge in a Neural Network, Hinton et al. 2015.

---

> ### Author Response · Authors · 2025-07-29
> **Response to Review**
>
> We thank the reviewer for their valuable and positive feedback. We address their concerns below:
>
> **Generalization:** Our approach prioritizes task-specific performance, which may compromise the generalization capabilities of the underlying models, particularly for architectures like DINO, whose backbones are broadly adopted across tasks. We will clearly acknowledge this as a limitation in the final version of the paper. Nonetheless, several backbones studied in our paper are already/inherently task-specific (e.g., saliency prediction), and our method provides clear and substantial benefits.
>
> **Bi-distillation instead of joint collaborative training:** Thank you for the suggestion. We will center the final narrative on bi-distillation as the core contribution.
>
>
> **Analysis of training dynamics and post-training analysis:** Thank you for the valuable suggestion. We observe an improvement in CCA between features after Bi-KD. For ViT-S and ViT-T, the mean CCA over the top 50 components increases from 0.7530 to 0.8133, computed over the CLS embeddings from the layer preceding the MLP head. Considering the top 100 components, the mean CCA improves from 0.6768 to 0.7487, post Bi-KD. We will include a detailed analysis across architectures in the final version of the paper.
>
> **Other training dynamics:** The model exhibiting higher confidence in the ground-truth label is designated as the teacher. This gives rise to three possible scenarios:
>
> * Case 1: Both the teacher and the student predict the correct class.
> * Case 2: The teacher predicts the correct class, while the student predicts incorrectly.
> * Case 3: Both the teacher and the student predict incorrectly, but the teacher assigns higher confidence to the ground-truth class than the student.
>
> In Cases 1 and 2, the student clearly benefits from distillation. Although Case 3 may negatively impact training, it is worth noting that the teacher still demonstrates better alignment with the ground truth. Additionally, the task-specific loss function contributes to training stability and mitigates the potential adverse effects of Case 3.
>
> At the start of training with ViT-S and ViT-T, Case 1 accounts for 70.9%, Case 2 for 12.4%, and Case 3 for 16.6%. Post Bi-KD, Case 1 rises to 73.1%, Case 2 drops to 10.1%, while Case 3 remains at similar levels, suggesting limited knowledge transfer in Case 3. We will add this analysis to the final version of the paper.
>
> **Training and implementation details:**
> - Batch size: 128
> - Epochs: 20
> - Augmentation: none for all models except ViTs, which use default augmentations from the TIMM library
> - Training: full precision
>
> These details will be included in the final version. We are also committed to open-source the code upon acceptance.

---

> ### Author Response · Authors · 2025-08-11
> **Response to Review**
>
> Dear reviewer XJcU, thank you for your time in going over our response. We would be happy to provide additional clarification or details if necessary over the last week of the discussion phase.

---

> > ### Comment · Reviewer_XJcU · 2025-08-11
> >
> > Dear authors,
> >
> > My issues were partially resolved, however there has been no revisions to the manuscript as far as I can tell. In your response, you stated you were to include some of the discussed improvements to the manuscript. My remaining concerns regard reframing of the narrative, as well as concerns on the lack of additional tasks to show the models are generalising, and not just overfitting on the particular task.
> >
> > I appreciate the included discussion on the training dynamics, as well as reporting of training details (although a little sparse, you should probably mention use or non-use of schedulers, LLRD, other details in the final manuscript), and I consider these concerns as resolved.

---

> > > ### Author Response · Authors · 2025-08-12
> > > **Response to Review**
> > >
> > > Dear reviewer XJcU, we have revised the manuscript to address the concerns raised in the review. We welcome any additional feedback or questions that you feel should be addressed to further strengthen the work.

---

> > > > ### Comment · Reviewer_XJcU · 2025-08-12
> > > >
> > > > Dear authors,
> > > >
> > > > I am pleased to see you take action and commit to improving your manuscript. I have updated my final recommendation to reflect this.
> > > >
> > > > I admittedly have one remaining concern on the generalisation of the method, and would have liked to see the method evaluated on additional (smaller) datasets. I mentioned this in the review as well. However, I believe this to be a minor concern at this point.

---

### Decision · Action_Editor_Pz3Z · 2025-08-20

**Recommendation:** Accept with minor revision

**Additional Comments:**

This paper presents a knowledge distillation method where multiple pretrained models collaborate to transfer knowledge among themselves. Teaching roles are dynamically assigned to models based on their prediction confidence for the ground-truth class, allowing each model to benefit from the strengths of others. The approach was validated across various computer vision tasks.

In the initial reviews, concerns were raised about generalization capability, the focus on supervised settings, a need to analyze training dynamics, missing technical and experimental details, insufficient explanation for the design choices, and marginal performance improvements, among others. The authors have largely addressed these issues in their responses and the revisions to the manuscript. This led to two reviewers leaning to accept. On the other hand, the other reviewer is leaning to reject for reasons that include marginal performance improvements and lack of statistical confidence levels.

The AE considers the concerns of the leaning-to-reject reviewer to be valid but notes that marginal performance improvement is not considered grounds for rejection in TMLR (but rather whether the claims are adequately substantiated and that there exists an audience for this work), and statistical confidence levels could be added in a minor revision. A decision was therefore made to accept the paper with minor revisions, where at least Table 4 in the final manuscript must report confidence levels, given the relatively small improvements shown in this experiment.

**Audience:**

Yes

**Audience Explanation:**

The AE and reviewers concur that the findings of this paper should be of interest to the TMLR audience.

**Claims And Evidence:**

Yes

**Claims Explanation:**

After the revisions made during the discussion period, the paper now provides sufficient evidence to support its claims. All the reviewers and the AE agree upon this.

---

> ### Author Response · Authors · 2025-09-01
> **Response to Decision**
>
> Dear Action Editor,
> Thank you for taking the time to assess our work and the overall positive decision. In line with your suggestion regarding confidence levels, we have rerun all experiments in Table 4 five times and have updated the manuscript accordingly, observing consistent improvements across runs.
> We are also in the process of rerunning all experiments in Table 2 three times. Owing to our limited computational resources, this will require a few additional days. We will inform you as soon as the revised results are incorporated.

---

> > ### Comment · Action_Editor_Pz3Z · 2025-09-03
> >
> > Dear Authors,
> >
> > Thank you for adding confidence levels to Table 4 as requested, and also for going further to add them to Table 2 as well. I look forward to seeing the revised results when they are ready.

---

> > > ### Author Response · Authors · 2025-09-18
> > > **Response to Decision**
> > >
> > > Dear Action Editor,
> > >
> > > We have submitted the camera-ready version of the paper, which now includes the confidence levels for Table 2.